# Biochemical and Physical Performance Responses to a Soccer Match after a 72-Hour Recovery Period

**DOI:** 10.3390/sports10100140

**Published:** 2022-09-22

**Authors:** Diego Marqués-Jiménez, Julio Calleja-González, Iñaki Arratibel-Imaz, Nicolás Terrados

**Affiliations:** 1Department of Didactics of Musical, Plastic and Corporal Expression, Faculty of Education, University of Valladolid, 42004 Soria, Spain; 2Department of Physical Education and Sports, Faculty of Education & Sport, University of the Basque Country (UPV/EHU), 01007 Vitoria-Gasteiz, Spain; 3Sports Medicine Center Tolosa Kirol Medikuntza, 20400 Tolosa, Spain; 4Regional Unit of Sports Medicine of Principado de Asturias, Municipal Sports Foundation of Avilés, 33401 Avilés, Spain; 5Health Research Institute of the Principality of Asturias (ISPA), 33011 Oviedo, Spain

**Keywords:** fatigue, hormone, physical performance, recovery, soccer

## Abstract

The physiological and neuromuscular responses at 72 h post-match are not widely researched, despite evidence showing substantial changes in recovery markers at 72 h post-match. Consequently, the aim of this study was to determine the biochemical and physical performance responses to a soccer match after a 72-h recovery period. Male soccer players of a semiprofessional team participated in this study. Before playing a friendly match, blood values of testosterone, cortisol, the testosterone-to-cortisol ratio and urea were collected and the squat jump and the Bangsbo Repeated Sprint Ability test were performed. These measurements were considered as baseline (pre match) and were obtained again after a 72-h recovery period. Results indicate that physical performance at 72 h post-match was similar to baseline (squat jump: *p* = 0.974; total Repeated Sprint Ability time: *p* = 0.381; Repeated Sprint Ability fatigue index: *p* = 0.864). However, perturbations in the biochemical milieu derived from the soccer match metabolic and physiological stress were still evident at this time point. While no significant differences compared to pre match were obtained in testosterone and urea concentrations after the recovery period, cortisol and testosterone-to-cortisol ratio values were significantly higher (14.74 ± 3.68 µg/dL vs. 17.83 ± 2.65 µg/dL; *p* = 0.045; ES 0.92 [0.00; 1.84], very likely) and lower (39.08 ± 13.26 vs. 28.29 ± 7.45; *p* = 0.038; ES −0.96 [−1.89; −0.04], very likely), respectively. In conclusion, soccer players have similar physical performance to the pre match after a 72-h recovery period, even with signs of biochemical and physiological stress.

## 1. Introduction

The activity of most soccer players during the competitive season entails 1-week cycles of training, tapering, competition and recovery. In some cases, players have to play two or three matches in just seven days. In this scenario, recovery kinetics and inflammatory adaptations in response to a three-match microcycle may display a different pattern, with strong indications of greater physiological stress and fatigue after matches preceded by only a 3-day recovery period [1]. An insufficient recovery time between soccer matches or an excessive training load prescription while players are recovering from match-related demands could reduce the player’s ability to positively adapt, or worse, injury and illness risk may be increased [2,3,4]. Therefore, one of the greatest challenges faced by coaches and athletes is to achieve an optimum balance between match-derived biochemical and physiological stress and recovery and to stimulate functional responses which positively affect physical performance [3,4].

Several biomarkers reflect the metabolic and physiological mechanisms underlying physical stress in soccer [5]. Serum creatine kinase (CK), myoglobin and lactate dehydrogenase (LDH) have been the standard biomarkers of muscle damage for a number of years, and interleukin-6 (IL-6), C-reactive protein (CRP) and tumor necrosis factor-alpha (TNF-α) are considered specific inflammatory markers [5]. Moreover, hormonal parameters could be useful to monitor fatigue and recovery, even for short time-periods [6]. However, testosterone (T) and cortisol (C) responses after a single soccer match and during the recovery period presents a high variability. Regarding T, both decrements [6,7] and increments [8] from pre match to the 0–30 min window post-match have been reported. Diminished T concentrations after 24 h and 48 h post-match have also been found [6]. Similarly, the 0–30 min post-match C levels may not change with respect to pre match [6,7,8], but if it increases, baseline levels may be restored between 24 h [9] and 72 h post-match [10]. Even when no pre- and post-match differences have been found, C concentrations may be diminished after 24 h and 48 h [6]. While C can be used as an indicator of physiological stress and a catabolic marker, the testosterone-to-cortisol ratio (T/C) is used as an indicator of anabolic/catabolic stress, fatigue and overreaching [11]. Regarding the T/C ratio, no differences have been identified from pre match to post match [8], but it has also been found that it may decrease by 64.2% after a match [7]. The T/C ratio concentrations may be still decreased at 48 h post-match [10].

This high variability in hormonal response, due to players’ individual characteristics and different metabolic and physiological responses derived from a soccer match, encourages sport scientists and physicians to measure several biomarkers together to monitor the recovery process in soccer [5]. Identifying the urea concentration can also provide important information. As urea is directly associated with the adaptation of protein metabolism to exercise, a pronounced increase indicates strong influence of both intensity and duration of exercise, whereas the normalization of the blood level is an indicator of recovery from fatigue [12,13]. Notwithstanding, different responses have also been observed after a soccer match. Immediately post-match, urea may increase, indicating that its metabolism is performed quickly [14]. However, a stable post-match urea level was also reported [15]. At any case, more rest is needed if the 24 h postexercise blood urea has not returned to the baseline concentration [12].

Biomarkers should be considered as an option to complement the information obtained through physical performance and perceptive markers [5,16], as both are often reported acutely impaired and exacerbated in the days post-match [17]. In this context, the match impact on player’s ability to jump or repeatedly perform sprint actions can be useful for recovery monitoring. This match-induced effect has been previously investigated with different jump and Repeated Sprint Ability (RSA) protocols. On the one hand, a meta-analysis revealed that match-induced fatigue impairs jump performance until 72 h post-match, with a small but consistent effect [17]. However, squat jump (SJ) and countermovement jump (CMJ) performance may have different recovery kinetics after soccer matches [18] due to the fatigue influence on neuromuscular parameters during the stretch-shortening cycle (SSC) [19]. On the other hand, RSA performance impairments in mean sprint time and fatigue indexes have been identified at post-match [20,21,22], but performance can be similar to baseline between 36 h [20] and 72 h post-match [23].

The abovementioned information demonstrates that the extent of the recovery period post-match in soccer players cannot consist of a ‘one size fits all approach’ [17]. Thus, there is a need for research to thoroughly understand the complexity of soccer players’ recovery via physiological and neuromuscular responses to match play. While most studies have investigated the recovery kinetics throughout the first days of recovery after a soccer match (24 h and 48 h), the physiological and neuromuscular responses at 72 h post-match are not as widely researched, despite evidence showing substantial changes in recovery markers at 72 h post-match [17]. This is particularly important because 72 h post-match is a key time-point where the next match or a hard training session may take place. In fact, the recommendation of the International Olympic Committee consensus, which indicated that soccer matches should be interspersed by at least 96 h [3,4], has still not been taken in consideration by different sports governing bodies, which do not allow longer recovery between official matches than 72 h [24,25]. Consequently, the objective of this study was to determine the biochemical and physical performance responses to a soccer match after a 72-h recovery period. The hypothesis of the present study is that the recovery of physical performance might deviate from biochemical responses.

## 2. Materials and Methods

### 2.1. Participants

Ten male soccer players (22.92 ± 2.38 years, 176.5 ± 4.84 cm, 71.18 ± 5.7 kg) of a semiprofessional team participated in this study. They received regular payment from their team, but at a considerably lower rate than a full-time professional player (they had a half-time employment). All participants were healthy, without serious musculoskeletal, metabolic, cardiovascular, respiratory, hematological or endocrine exercise disorders in the previous 6 months. Players were informed about the objectives, risks and discomforts of the study and gave their informed consent for participation. This study respects the Declaration of Helsinki (1964) and was approved by the Ethics Committee of the University of the Basque Country (CEISH/357/2015/MARQUÉS JIMÉNEZ).

### 2.2. Research Design

All the procedures were carried out in-season, concretely during the winter break (week without official competition). Participants were not involved in any type of training 48 h before baseline measurements. This washout period served to minimize the influence of fatigue-induced carryover effects experienced by the players within the in-season schedule. They were also advised to avoid any type of physical activity during the 72-h recovery period.

One hour before the friendly match (9:00 am), and after an overnight fast, anthropometric data and plasma values of T, C, the T/C ratio and urea were collected. After that, participants were required to perform 3 SJ, with a 3 min recovery between attempts, and one RSA test. These measurements were considered as baseline (pre match) and were obtained again at 72 h post-match. Instructions of each test were provided three weeks before baseline measurements during a familiarization session.

A standard warm-up was conducted after the blood sample data collection and before the physical performance tests. Warm-up included 8 min of light jogging to increase core temperature, dynamic stretching, coordination, jumps, change of direction drills, acceleration runs and 2 single 2 × 15 m sprints with 1 min of passive recovery in-between. A 3 min pause was given before players undertook the SJ and RSA tests. 

The soccer match lasted 90 min, and each participant played the full time, as no substitutions were permitted. The soccer match was played in the morning (10:00 am) on an artificial turf pitch, but data collection was obtained in a covered pavilion, where environmental conditions were similar in both baseline and 72 h post-match measurements. All measurements were performed at the same time of the day to avoid circadian variation.

Participants had to refrain from taking anti-inflammatory drugs, nutritional supplements or other prescription drugs and from any plausible recovery strategy for 7 days before the study and during it. Participants were also advised to abstain from consuming alcohol or caffeine 24 h before baseline measurements and during the study. Players recorded their dietary intake 24 h before baseline measurement and had to replicate it every day, except breakfast after the blood samples, which was standardized.

### 2.3. Blood Sampling

Blood samples were obtained with antecubital vein puncture according to standard diagnostic procedures. All blood samples were centrifuged at 3000 rpm for 10 min, and the serum of each sample was immediately frozen at −80 °C for later biochemical analysis. All samples were handled within a single laboratory. The C, T and T/C ratio values were analyzed using an Auto Chemistry Analyzer BM-100 (BioMaxima S.A., Lublin, Poland). The mass spectrometry method was utilized to measure hormones. The analyzer used was maintained by regular quality control procedures according to the manufacturer’s instructions to avoid any inconvenience during the procedures. Urea was analyzed with standard routine measurements by using an ERBA Chem 7 analyzer (Erba Mannheim, Mannheim, Germany). 

### 2.4. Physical Performance Measurements

SJ height was measured by Optojump Next (Microgate^®^, Bolzano, Italy), and mean SJ values of the 3 attempts were considered for statistical analysis. The protocol of Bosco, Luhtanen and Komi [26] was used and required participants to perform the SJ from a half-squat position, with knees bent at 90°, torso straight and both hands on their waist.

The RSA performance was evaluated using the 7 × 34.2 m Bangsbo RSA test [27]. The photocell gates (Racetime2^®^, Microgate, Bolzano, Italy) were placed 0.4 m above the ground at the start and at 34.2 m. Only one attempt of the RSA test was performed. When ready, the players started the test from a standing position 0.5 m away from the first photocell gate and sprinted a total distance of 34.2 m, involving a swing after the first 10 m, then continued the sprint to the finish line where a photocell was placed (finish time). Next, they performed a 25 s active recovery of jogging on the return to the starting line. The 7 × 34.2 m repeated sprint was performed as fast as possible, and verbal feedback was provided to each player during the recovery period in order to facilitate the readiness for the next sprint on time. The test finished when each player completed the 7 × 34.2 m repeated sprints. The following variables were calculated: (a) total RSA time (RSAt), the sum of the seven sprint times; (b) the fatigue index (RSAfi), calculated using the following formula [100 × (TT/(BT × 10)) − 100], where TT corresponds to RSAt and BT to the best time [28].

### 2.5. Match Loads Monitoring

External and internal match loads were also monitored using a short-range telemetry and triaxial accelerometer incorporated within the GPS (Polar Team Pro, Polar Electro^®^, Kempele, Finland). Internal loads metrics recorded were: average heart rate (HR) during the match calculated as a percentage of the maximal HR (%HRavg), maximal HR during the match calculated as a percentage of the maximal HR (%HRmax) and time spent (min) at various intensities expressed as a percentage of HRmax: 50–59%, 60–69%, 70–79%, 80–89% and >90% of HR_max_. External loads metrics monitored were: total distance covered (TD), distance at various speed thresholds (3.00 to 6.99 km/h, 7.00–10.99 km/h, 11.00–14.99 km/h, 15.00–18.99 km/h and >19.00 km/h), number of sprints (>23 km/h), number of low-intensity (<1.0 m/s^2^), low-to-moderate-intensity (1.0 to 1.9 m/s^2^), moderate-to-high-intensity (2.0 to 2.9 m/s^2^) and high-intensity (>3.0 m/s^2^) accelerations, and number of low-intensity (≤1.0 m/s^2^), low-to-moderate-intensity (≤1.0 to −1.9 m/s^2^), moderate-to-high-intensity (≤2.0 to −2.9 m/s^2^) and high-intensity (≥3.0 m/s^2^) decelerations.

### 2.6. Statistical Analyses

Descriptive statistics are reported as means and standard deviations (M ± SD). Between-subject reliability of internal and external match load variables was assessed using the percentage of coefficient of variation (CV%). The intraclass correlation coefficient (ICC), along with the upper and lower 95% confidence interval (CI), was used to determine the relative between-day reliability of each measurement. The Shapiro–Wilk test was applied to evaluate whether the data were normally distributed and the Levene test was used to evaluate the homogeneity of variances. Differences between pre- and 72 h post-match were compared using the paired-samples *t*-test and Cohen’s effect size (ES). Threshold values for standardized differences were <0.2 (trivial), 0.2–0.5 (small), 0.5–0.8 (moderate) and >0.8 (large) [29]. A qualitative probabilistic mechanistic inference (90% confidence intervals) was applied, with inferences based on standardized thresholds for the smallest worthwhile change (SWC), which was set as 0.2 of baseline SD [30]. The qualitative probabilistic terms were assigned using the following scale [31]: <0.5%, most unlikely or almost certainly not; 0.5–5%, very unlikely; 5–25%, unlikely or probably not; 25–75%, possibly; 75–95%, likely or probably; 95–99.5%, very likely; >99.5%, most likely or almost certainly. Pearson correlation was used to measure the degree of association between match-related loads and percentage change (delta; Δ) between pre- and 72 h post-match measurements. It was calculated with variables which reached statistical significance when comparing pre- and 72 h post-match. The following thresholds were considered to interpret the correlation coefficient [30]: trivial (≤0.1), small (>0.1–0.3), moderate (>0.3–0.5), large (>0.5–0.7), very large (>0.7–0.9) and almost perfect (and >0.9–1.0). Statistical significance was inferred at *p* < 0.05. Statistical analyses were carried out by SPSS 20.0 software (SPSS^®^, Chicago, IL, USA).

## 3. Results

Match-related loads (both internal and external) are presented in Table 1.

Figure 1 represents individual changes of each biomarker between pre- and 72 h post-match and Figure 2 shows standardized differences in biomarkers responses. Based on the ICC results, the between-day relative reliability was excellent for T (0.93; 0.70, 0.98), and good for C (0.74; −0.04, 0.94), T/C (0.72; −0.14, 0.93) and urea (0.72; −0.15, 0.93). No significant differences were obtained in T (5.33 ± 0.78 ng/dL vs. 4.90 ± 0.86 ng/dL; *p* = 0.262) and urea concentrations after the recovery period (37.00 ± 4.55 mg/dL vs. 34.40 ± 4.90 mg/dL; *p* < 0.235), but both T (ES −0.50, [−1.39;0.39], likely) and urea (ES −0.53, [−1.42;0.36], likely) concentrations were moderately lower. At 72 h post-match, C and T/C values were largely higher (14.74 ± 3.68 µg/dL vs. 17.83 ± 2.65 µg/dL; *p* = 0.045; ES 0.92 [0.00;1.84], very likely) and lower (39.08 ± 13.26 vs. 28.29 ± 7.45; *p* = 0.038; ES −0.96 [−1.89; −0.04], very likely) compared to pre match values, respectively.

Table 2 represents mean differences, standardized differences and qualitative probabilistic mechanistic inference of physical performance responses. SJ performance and RSA-derived variables showed no significant differences between pre- and 72 h post-match (*p* < 0.05). ES of SJ and RSAfi was possibly trivial, but a possibly small ES was obtained after recovery period in RSAt.

Table 3 provides significant correlations between independent and dependent variables. On one hand, a very large correlation was found between the ΔC and ΔT/C ratio, whereas urea was largely correlated with the ΔC and ΔT/C ratio. On the other hand, large and very large correlations between time spent at various intensities, expressed as the percentage of HR_max_ and the ΔC and ΔT/C ratio, were also found. However, external loads did not correlate with measurements, which showed statistical difference between the time-points.

## 4. Discussion

The objective of this study was to determine the biochemical and physical performance responses to a soccer match after a 72-h recovery period. Results show that physical performance at 72 h post-match was similar to baseline. However, perturbations in the biochemical milieu derived from the soccer match metabolic and physiological stress were still evident at this time point. While no significant differences compared to rematch were obtained in T and urea concentrations after a 72-h recovery period, the C and T/C values were significantly higher and lower, respectively. These results contrast with those previously reported in a brilliant systematic review [17]. The authors indicated that hormone concentrations may be fully restored after a recovery period of 72 h post-match, but this may not be long enough to completely recover homeostatic balance (muscle damage, physical and well-being status). These conflicting results highlight the high interindividual variability in acute fatigue and recovery processes in soccer players, as have been previously reviewed [5,16]. 

Regarding T, previous studies have reported contradictory results at post-match, as T can be decreased [6,7], increased [8] or even unchanged [9]. Focused on recovery, a meta-analysis indicated that a soccer match may not alter the T level during the recovery period [32], but diminished T concentrations after 24 h (−25%) and 48 h post-match (−30%) have also been previously found [6]. Our results show that T concentrations were moderately lower at 72 h post-match, although these decrements do not reach statistical significance. Consequently, it seems that a 72-h recovery period may be long enough to restore baseline levels. Two main factors may be considered to explain discrepancies between the present findings and the opposed ones. On one hand, the absence of significant differences may be influenced by the competitive level of participants, because high-level players have higher T reactivity compared to novice players [32]. On the other hand, the match of this study was friendly, so the moderately but nonsignificant decrements on T levels after 72 h post-match cannot be linked to an additional metabolic stress derived from official matches, whose situational variables (e.g., match result, match location and strength of team and opponent) may lead to different responses in T concentrations [11,32].

C is the main glucocorticoid secreted in response to physical and psychological stress and may be responsible for the catabolic effects of exercise [33]. In soccer, different studies indicated that post-match C levels may not change with respect to pre match [6,7,8]. However, plasma C concentration may be significantly increased at 24 h and 48 h compared to baseline [32]. Our findings indicated that C concentration was significantly and largely increased after a 72-h recovery period. This suggest that participants were involved in a negative balance between biochemical and physiological stress and recovery, as C response may induce lower T secretion [34], contributing therefore to the catabolic process. The different time-course of recovery in C response reported in previous studies [9,10] with respect to our results could be explained by two interactive factors: competitive soccer matches increase C levels to a greater magnitude compared to noncompetitive fixtures, and male novice players show greater levels of C reactivity compared to high-level soccer players [32].

The T/C ratio can be used as an indicator of the relationship between anabolism and catabolism. The reduction of this ratio would indicate a predominance of catabolic processes, whilst an increase would indicate a predominance of anabolic processes [11]. Some studies have identified no differences in T/C ratio between pre- to post match [6,8] or throughout a 48-h recovery period [6]. Other authors have reported that T/C ratio can decrease by 64.2% after the match [7], and may be significantly lower after 24 h and 48 h post-match before to return to baseline levels at 72 h post-match [10]. Our findings show that T/C ratio was significantly and largely decreased after a 72-h recovery period. The hypothesis that T/C may not return to baseline levels after 72 h due to a high pre match value should be discard. Considering that C response may induce lower T secretion [34], the correlation found between the ΔC and T/C ratio suggests that C is involved in a transient negative response to the metabolic and physiological stress derived from a soccer match.

Conflicting results have been previously obtained in the relationship between match-loads and hormonal responses after soccer matches. Peñailillo et al. [7] reported that TD correlated with changes in T concentrations from pre- to post-match (r = 0.85; *p* = 0.004), but Thorpe and Sunderland [8] did not find correlations between match activity metrics and the T, C or the T/C ratio at post-match. During recovery period, Romagnoli et al. [6] showed that TD correlated with the values for post-24 h C (r = 0.502, *p* = 0.034) and the increase in C at 48 h with respect to pre-match values (r = 0.515, *p* = 0.029). In contrast, we found that the Δ of hormone concentrations correlated exclusively with some internal load variables (time at 50–59% HR_max_ and time at 80–89% HR_max_). These relationships may indicate the association between the internal load and the match-induced metabolic response. Differences among studies may be linked to the high interindividual variability of players, competitive level and match requirements [5,16,32].

The urea plasma level is commonly used to assess protein catabolism and purine nucleotides degradation [35], but it may also be suitable for exercise-related stress measurement [13]. In this context, urea concentrations significantly higher than baseline at 24 h postexercise may indicate insufficient recovery [12]. Our findings show that urea concentrations were not significantly different at 72 h compared to baseline, although ES indicates that values were moderately lower at this point. This suggest a probable compensation mechanism of energy sources during the 72-h recovery period, which could be connected with protein catabolism and purine nucleotides metabolism. However, this hypothesis should be analyzed with caution, because urea variations may also be influenced by other factors, such as protein intake, state of hydration, hepatic urea synthesis or renal urea excretion [36].

Our results show no significant differences between pre- and 72 h post-match on neuromuscular (decrease in force production and power related to SJ height) or physical performance impairments (RSA-derived indices decrements). On one hand, the match-induced fatigue impairs jump performance until 72 h post-match [17]. However, our results indicate that SJ height at 72 h post-match was similar to baseline. In fact, SJ performance after soccer matches seems to recover earlier than CMJ [18] because SJ does not involve SSC, which may need 4–8 days to recover depending on the severity of exercise [19]. On the other hand, we found that RSA performance at 72 h post-match was similar to baseline, similarly to previous findings [23]. Notwithstanding, RSA performance could have return to baseline before, as have been reported [20]. Although we did not measure SJ or RSA performance immediately postmatch, the hypothesis of an insufficient match-induced fatigue to deteriorate both performance measurements seem unlikely. Previous studies showed that match-induced fatigue is reflected in significant jump decrements [17] and in RSA performance impairments [20,21,22]. Moreover, ES of match-induced fatigue on SJ and RSAfi after recovery period were trivial while in RSAt were small. This may indicate the presence of match-induced fatigue during the recovery period, which may affect distinctly SJ and RSA performance. It is likely that recovery kinetics between SJ and RSA performance were different, because match-induced biomechanical (e.g., force production and eccentric and concentric actions) and metabolic (e.g., glycogen stores) impairments may be distinctly affected [37,38,39]. 

From a practical point of view, as well as considering that the activity of most soccer players during the competitive season does not include 3 days off after matches, our results suggest that coaching staffs should adjust the training workload, structure and content during the 72 h post-match training sessions. These actions let players achieve an optimum balance between matches-derived biochemical and physiological stress and recovery. If players and coaches do not sufficiently respect the balance between workload and recovery, nonfunctional overreaching could occur. Moreover, practitioners should make training sessions or friendly matches as similar to competition as possible to simultaneously solicit the endocrine system, decrease the psycho-physiological stress and improve the recovery pattern in highly competitive scenarios.

The study’s novelty is that some measurements that are not typically evaluated in previous studies regarding this topic were measured, such as SJ and RSA. Moreover, physiological and neuromuscular responses at 72 h post-match is not as widely researched, so this study contributes to a deeply understanding of the recovery process after a soccer match. However, the following limitations should be indicated. The small number of study participants is a limitation of the current study. The results could be different in professional players or teams who played two- or three-match by week. Although players were not involved in any type of training 48 h before baseline measurements, the pre- match measures of the C, T and urea may be impacted by the prior training these players performed. Moreover, utilizing an outcome metric such as jump height alone may mask the effects of fatigue. Some athletes may alter their jump mechanics when fatigued in order to help maintain jump height [37]. Recent recommendations have suggested that the ratio of flight time to contraction time (FT: CT) may be a more sensitive measure of recovery [40]. However, equipment availability did not allow us to measure it. Future studies should be directed not only to the study of the impact of soccer matches on football players of different levels and gender, but also on soccer players of different ages and friendly and competitive games.

## 5. Conclusions

In summary, this study shows that physical performance at 72 h post-match was similar to baseline. However, perturbations in the biochemical milieu derived from the soccer match metabolic and physiological stress were still evident at this time point. While no significant differences compared to pre match were obtained in the T and urea concentrations after a 72-h recovery period, the C and T/C values were significantly higher and lower, respectively. Consequently, semiprofessional soccer players may have similar physical performance after a 72-h recovery period with respect to pre match, even with signs of biochemical and physiological stress. These findings confirm that the recovery of physical performance might deviate from biochemical responses.

## Figures and Tables

**Figure 1 sports-10-00140-f001:**
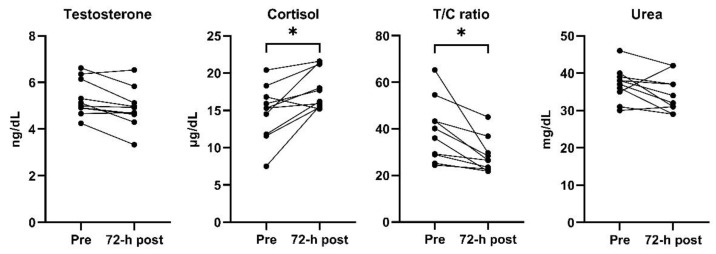
Individual changes of biomarkers between pre- and 72 h postmatch. * Statistically significant difference (*p* < 0.05).

**Figure 2 sports-10-00140-f002:**
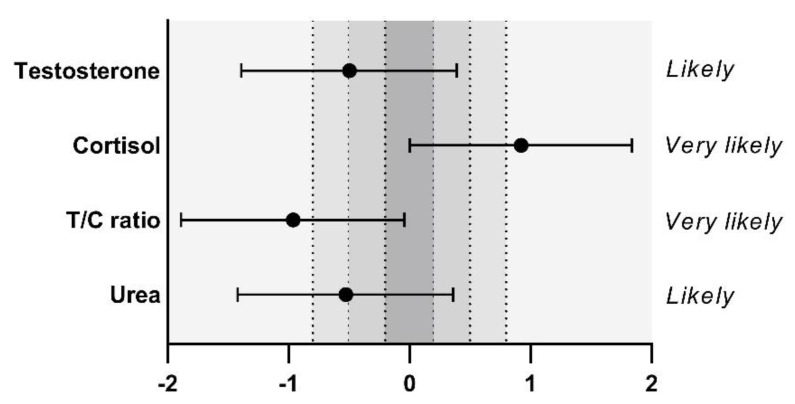
Standardized differences in biomarkers responses after a 72 h recovery period and qualitative probabilistic mechanistic inference.

**Table 1 sports-10-00140-t001:** Match-related internal and external loads during the match.

	M + SD	CV (%)
Internal loads		
%HRavg	77.00 ± 6.36	8.3
%HRmax	95.30 ± 5.68	6.0
Time (min) at 50–59% HRmax	4.03 ± 5.37	102.3
Time (min) at 60–69% HRmax	15.06 ± 10.05	66.8
Time (min) at 70–79% HRmax	27.89 ± 9.13	32.7
Time (min) at 80–89% HRmax	32.41 ± 10.47	32.3
Time (min) at 90–100% HRmax	10.62 ± 14.29	103.1
External loads		
TD (m)	9890.55 ± 684.85	6.9
TD (m) at 3.00–6.99 km/h	4038.12 ± 329.33	8.2
TD (m) at 7.00–10.99 km/h	2964.03 ± 376.20	12.7
TD (m) at 11.00–14.99 km/h	1787.13 ± 454.91	25.5
TD (m) at 15.00–18.99 km/h	758.08 ± 197.39	26
TD (m) at >19.00 km/h	343.19 ± 148.14	43.2
No. sprints (>23 km/h)	9.80 ± 4.66	47.6
No. high-intensity decelerations (>−3.0 m/s^2^)	65.70 ± 9.43	14.4
No. moderate-to-high-intensity decelerations (−2.9 to −2.0 m/s^2^)	131.20 ± 24.14	18.4
No. low-to-moderate decelerations (−1.9 to −1.0 m/s^2^)	159.80 ± 24.86	15.6
No. low decelerations (−0.9 to −0.5 m/s^2^)	415.30 ± 24.73	6
No. low accelerations (0.5 to 0.9 m/s^2^)	394.80 ± 33.38	8.5
No. low-to-moderate accelerations (1.0 to 1.9 m/s^2^)	236.20 ± 44.51	18.8
No. moderate-to-high-intensity accelerations (2.0 to 2.9 m/s^2^)	130.40 ± 14.55	11.2
No. high-intensity accelerations (>3.0 m/s^2^)	62.00 ± 10.77	17.4

CV: coefficient of variation; HRmax: maximum heart rate; HRavg: average heart rate; TD: total distance covered.

**Table 2 sports-10-00140-t002:** Mean differences (M ± SD), intraclass correlation coefficient (ICC), standardized differences (ES) and qualitative probabilistic mechanistic inference of physical performance measurements between pre- and 72 h post-match.

	SJ Height (cm)	RSAt (s)	RSAfi (%)
Pre match	30.11 ± 4.53	43.52 ± 2.12	3.49 ± 0.99
72 h post-match	30.17 ± 4.61	44.37 ± 2.11	3.66 ± 2.36
*p*	0.974	0.381	0.864
ICC (95% CI)	0.97 (0.88, 0.99)	0.79 (0.13, 0.95)	0.64 (−0.44, 0.91)
ES (95% CI)	0.01 (−0.86, 0.89)	0.38 (−0.50, 1.27)	0.09 (−0.79, 0.97)
ES magnitude	Trivial	Small	Trivial
Probabilistic inference	Possibly	Possibly	Possibly

CI: confidence interval; ICC: intraclass correlation coefficient; ES: effect size; RSAfi: Repeated Sprint Ability fatigue index; RSAt: Total Repeated Sprint Ability time; SJ: squat jump.

**Table 3 sports-10-00140-t003:** Correlations between match-related loads and percentage change (Δ) of biomarkers.

	r	r Magnitude	*p*
Biomarkers (Δ)			
ΔC and ΔT/C	−0.881	very large	0.001
ΔC and Δurea	0.693	moderate	0.026
ΔT/C ratio and Δurea	−0.679	moderate	0.031
Match-related loads and biomarkers (Δ)			
Time at 50–59% HRmax and ΔC	0.839	very large	0.002
Time at 50–59% HRmax and ΔT/C ratio	−0.644	moderate	0.045
Time at 80–89% HRmax and ΔC	−0.657	moderate	0.002

C: cortisol; HRmax: maximum heart rate; T/C: testosterone to cortisol ratio.

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
