# Peer review of "Biochemical and Physical Performance Responses to a Soccer Match after a 72-Hour Recovery Period"

_sports, 2022, doi:10.3390/sports10100140_

Round 1

Reviewer 1 Report

Review- Biochemical and physical performance responses to a soccer match after a 72-hours recovery period

First, I would like to recognize the authors for the data they analyzed to determine the biochemical and physical performance responses to a soccer match after a 72-h recovery period.

Abstract

The abstract provides an objective representation of the article and presents the study's results and significance without exaggerating.

Some minor suggestions

Line 13: Responses are…(minor issue). The singular verb “is” does not appear to agree with the plural subject “responses”.

Line 26: You use “functional states” referring to the players' physical performance. The term “functional states” is commonly used in psychology as well as when we refer to brain function. However, it is not used (based on my knowledge) when discussing physical performance. Therefore, I would suggest that you explain it differently.

Introduction

The introduction highlights the importance of the study. The study's purpose and significance are presented.

Some minor suggestions

Line 39: Therefore, greats challenges – Do you mean greatest? Or great? Please check

Line 50: with respect.

Line 66: you are correct. Urea and ammonia. This is not an issue is just a personal question. Did you check creatinine levels? (NOT an issue regarding the manuscript)

Methods

 The materials and methods section is presented with sufficient detail so that someone can replicate and build on the published results. In addition, the methods were appropriately presented and cited.

Based on your methodology, the players fasted overnight and then had their blood measurements and physical fitness tests one hour before the friendly game. Did they play without eating or drinking anything? I ask this because having a friendly game without consuming anything before the game (after overnight fasting) may have a different effect on metabolic, hormonal and physiological responses instead if they played the game after consuming the recommended carbohydrates etc.

Discussion

Lines 244-245: Regarding T, previous studies have reported contradictory results at post-match, as T can be decreased [6,7], increased [8] or even unchanged [9]. However, these findings are not relevant according to the aim of this study. So why do you include this if it is irrelevant to your study? I say this because the flow of this paragraph is not very good. A good critical analysis follows. Do not exclude the studies you included that tested T post-match; instead, find a better way to connect it to the subsequent sentences. Change only –however, these findings are not relevant according to the aim of this study- to something more precise.

Line 262: may not change respect to pre-match- please check your writing here (with respect instead of respect to pre-match).

Lines 268-270: “The differences between previous results [9,10] and ours could be explained by two interactive factors: competitive soccer matches increase C levels to a greater magnitude compared to non-competitive fixtures, and male novice players show greater levels of C reactivity compared to high-level soccer players [32]”. You also showed significant increases in C after 72 hours. Which differences are you referring to? Those that found no changes after post-match? (I guess you are not referring to those studies as you did not measure C post-match). Also, are you comparing your results to those that found increases in 24 and 48 h? Still, you did not measure C after 24 and 48 h. I know the study of Ispirlidis et al. (2008) and Silva et al. (2013) that you are citing; therefore, you may want to be more specific here as it is not clear when you say, “The differences between previous results [9,10] and ours could be explained….”

Line 287: failed to found- check your writing, please.

Line 294: “enabling precisely information about the metabolical stress experience by soccer players”- check your writing, please.

Line 311: fatigue may impairs jump – change the verb form to impair

Line 312: results indicates -change the verb form to indicate

Line 314: which reflects fatigue in a greater extent due inability – please check your writing

Limitations are presented, which is essential.

Author Response

Editor-in-Chief of the Sports:

This letter is in regard to the resubmission (first review) of our manuscript.  As a result of the peer review process in the first round and after considering all of the reviewer comments, we have revised our manuscript according to your suggestions.

We believe that your all comment and our corrections have significantly improved the article, and we hope that it is now suitable for publication in your respected journal.

In Advanced

Best Regards

Julio Calleja González

REVISOR 1

Some minor suggestions

Line 13: Responses are…(minor issue). The singular verb “is” does not appear to agree with the plural subject “responses”.

 It has been modified.

Line 26: You use “functional states” referring to the players' physical performance. The term “functional states” is commonly used in psychology as well as when we refer to brain function. However, it is not used (based on my knowledge) when discussing physical performance. Therefore, I would suggest that you explain it differently.

It has been modified.

Introduction

The introduction highlights the importance of the study. The study's purpose and significance are presented.

Some minor suggestions

Line 39: Therefore, greats challenges – Do you mean greatest? Or great? Please check

It has been modified.

Line 50: with respect.

It has been modified.

Line 66: you are correct. Urea and ammonia. This is not an issue is just a personal question. Did you check creatinine levels? (NOT an issue regarding the manuscript)

It was not measured.

Methods

 The materials and methods section is presented with sufficient detail so that someone can replicate and build on the published results. In addition, the methods were appropriately presented and cited.

Based on your methodology, the players fasted overnight and then had their blood measurements and physical fitness tests one hour before the friendly game. Did they play without eating or drinking anything? I ask this because having a friendly game without consuming anything before the game (after overnight fasting) may have a different effect on metabolic, hormonal and physiological responses instead if they played the game after consuming the recommended carbohydrates etc.

 In line 136 appears the following sentence: “except breakfast after blood samples, which was standardized”.

Discussion

Lines 244-245: Regarding T, previous studies have reported contradictory results at post-match, as T can be decreased [6,7], increased [8] or even unchanged [9]. However, these findings are not relevant according to the aim of this study. So why do you include this if it is irrelevant to your study? I say this because the flow of this paragraph is not very good. A good critical analysis follows. Do not exclude the studies you included that tested T post-match; instead, find a better way to connect it to the subsequent sentences. Change only –however, these findings are not relevant according to the aim of this study- to something more precise.

 It has been modified.

Line 262: may not change respect to pre-match- please check your writing here (with respect instead of respect to pre-match).

It has been modified.

Lines 268-270: “The differences between previous results [9,10] and ours could be explained by two interactive factors: competitive soccer matches increase C levels to a greater magnitude compared to non-competitive fixtures, and male novice players show greater levels of C reactivity compared to high-level soccer players [32]”. You also showed significant increases in C after 72 hours. Which differences are you referring to? Those that found no changes after post-match? (I guess you are not referring to those studies as you did not measure C post-match). Also, are you comparing your results to those that found increases in 24 and 48 h? Still, you did not measure C after 24 and 48 h. I know the study of Ispirlidis et al. (2008) and Silva et al. (2013) that you are citing; therefore, you may want to be more specific here as it is not clear when you say, “The differences between previous results [9,10] and ours could be explained….”

It has been modified.

Line 287: failed to found- check your writing, please.

It has been modified.

Line 294: “enabling precisely information about the metabolically stress experience by soccer players”- check your writing, please.

It has been modified.

Line 311: fatigue may impair jump – change the verb form to impair

It has been modified.

Line 312: results indicate -change the verb form to indicate

It has been modified.

Line 314: which reflects fatigue in a greater extent due inability – please check your writing

It has been modified.

Abstract

  • Lines 16-19: The authors provided the following method in the abstract:

“Before playing a friendly match, blood values of testosterone, cortisol, testosterone to cortisol ratio and urea were collected and squat jump and the Bangsbo Repeated Sprint Ability test were performed. These measurements were considered as baseline (pre-match), and were obtained again after a 72-h recovery period.”

  • This part needs to be rewritten entirely as it is vague and does not provide any information about the participants.

It has been modified.

Introduction

  • Lines 19-20: The exact p-value for the significant changes should be mentioned.

It has been modified.

  • Lines 39-42: Provide 3-5 references for the following sentence:

“Therefore, greats challenges faced by coaches and athletes are to achieve an optimum balance between match-derived biochemical and physiological stress and recovery and to stimulate functional responses which positively affects physical performance.

 References have been added.

  • Line 39: Change the word “greats” to “great”.

It has been modified.

  • Line 41: Change the word “affects” to “affect”.

 It has been modified.

  • The study hypothesis needs to be added at the end of "Introduction".

It has been modified.

Methods

  • Day-to-day test reliability, CV range, and intraclass correlation coefficients for the assessments must be included for ALL the assessments.

 It has been included in the results section (tables and text).

Discussion

  • The "Discussion" section needs to clarify the study's novelty.

It has been added.

  • The conclusions section is too brief.

 It has been expanded.

King Regards

Editor-in-Chief of the Sports:

This letter is in regard to the resubmission (first review) of our manuscript.  As a result of the peer review process in the first round and after considering all of the reviewer comments, we have revised our manuscript according to your suggestions.

We believe that your all comment and our corrections have significantly improved the article, and we hope that it is now suitable for publication in your respected journal.

In Advanced

Best Regards

Julio Calleja González

REVISOR 1

Some minor suggestions

Line 13: Responses are…(minor issue). The singular verb “is” does not appear to agree with the plural subject “responses”.

 It has been modified.

Line 26: You use “functional states” referring to the players' physical performance. The term “functional states” is commonly used in psychology as well as when we refer to brain function. However, it is not used (based on my knowledge) when discussing physical performance. Therefore, I would suggest that you explain it differently.

It has been modified.

Introduction

The introduction highlights the importance of the study. The study's purpose and significance are presented.

Some minor suggestions

Line 39: Therefore, greats challenges – Do you mean greatest? Or great? Please check

It has been modified.

Line 50: with respect.

It has been modified.

Line 66: you are correct. Urea and ammonia. This is not an issue is just a personal question. Did you check creatinine levels? (NOT an issue regarding the manuscript)

It was not measured.

Methods

 The materials and methods section is presented with sufficient detail so that someone can replicate and build on the published results. In addition, the methods were appropriately presented and cited.

Based on your methodology, the players fasted overnight and then had their blood measurements and physical fitness tests one hour before the friendly game. Did they play without eating or drinking anything? I ask this because having a friendly game without consuming anything before the game (after overnight fasting) may have a different effect on metabolic, hormonal and physiological responses instead if they played the game after consuming the recommended carbohydrates etc.

 In line 136 appears the following sentence: “except breakfast after blood samples, which was standardized”.

Discussion

Lines 244-245: Regarding T, previous studies have reported contradictory results at post-match, as T can be decreased [6,7], increased [8] or even unchanged [9]. However, these findings are not relevant according to the aim of this study. So why do you include this if it is irrelevant to your study? I say this because the flow of this paragraph is not very good. A good critical analysis follows. Do not exclude the studies you included that tested T post-match; instead, find a better way to connect it to the subsequent sentences. Change only –however, these findings are not relevant according to the aim of this study- to something more precise.

 It has been modified.

Line 262: may not change respect to pre-match- please check your writing here (with respect instead of respect to pre-match).

It has been modified.

Lines 268-270: “The differences between previous results [9,10] and ours could be explained by two interactive factors: competitive soccer matches increase C levels to a greater magnitude compared to non-competitive fixtures, and male novice players show greater levels of C reactivity compared to high-level soccer players [32]”. You also showed significant increases in C after 72 hours. Which differences are you referring to? Those that found no changes after post-match? (I guess you are not referring to those studies as you did not measure C post-match). Also, are you comparing your results to those that found increases in 24 and 48 h? Still, you did not measure C after 24 and 48 h. I know the study of Ispirlidis et al. (2008) and Silva et al. (2013) that you are citing; therefore, you may want to be more specific here as it is not clear when you say, “The differences between previous results [9,10] and ours could be explained….”

It has been modified.

Line 287: failed to found- check your writing, please.

It has been modified.

Line 294: “enabling precisely information about the metabolically stress experience by soccer players”- check your writing, please.

It has been modified.

Line 311: fatigue may impair jump – change the verb form to impair

It has been modified.

Line 312: results indicate -change the verb form to indicate

It has been modified.

Line 314: which reflects fatigue in a greater extent due inability – please check your writing

It has been modified.

Abstract

  • Lines 16-19: The authors provided the following method in the abstract:

“Before playing a friendly match, blood values of testosterone, cortisol, testosterone to cortisol ratio and urea were collected and squat jump and the Bangsbo Repeated Sprint Ability test were performed. These measurements were considered as baseline (pre-match), and were obtained again after a 72-h recovery period.”

  • This part needs to be rewritten entirely as it is vague and does not provide any information about the participants.

It has been modified.

Introduction

  • Lines 19-20: The exact p-value for the significant changes should be mentioned.

It has been modified.

  • Lines 39-42: Provide 3-5 references for the following sentence:

“Therefore, greats challenges faced by coaches and athletes are to achieve an optimum balance between match-derived biochemical and physiological stress and recovery and to stimulate functional responses which positively affects physical performance.

 References have been added.

  • Line 39: Change the word “greats” to “great”.

It has been modified.

  • Line 41: Change the word “affects” to “affect”.

 It has been modified.

  • The study hypothesis needs to be added at the end of "Introduction".

It has been modified.

Methods

  • Day-to-day test reliability, CV range, and intraclass correlation coefficients for the assessments must be included for ALL the assessments.

 It has been included in the results section (tables and text).

Discussion

  • The "Discussion" section needs to clarify the study's novelty.

It has been added.

  • The conclusions section is too brief.

 It has been expanded.

King Regards

Editor-in-Chief of the Sports:

This letter is in regard to the resubmission (first review) of our manuscript.  As a result of the peer review process in the first round and after considering all of the reviewer comments, we have revised our manuscript according to your suggestions.

We believe that your all comment and our corrections have significantly improved the article, and we hope that it is now suitable for publication in your respected journal.

In Advanced

Best Regards

Julio Calleja González

REVISOR 1

Some minor suggestions

Line 13: Responses are…(minor issue). The singular verb “is” does not appear to agree with the plural subject “responses”.

 It has been modified.

Line 26: You use “functional states” referring to the players' physical performance. The term “functional states” is commonly used in psychology as well as when we refer to brain function. However, it is not used (based on my knowledge) when discussing physical performance. Therefore, I would suggest that you explain it differently.

It has been modified.

Introduction

The introduction highlights the importance of the study. The study's purpose and significance are presented.

Some minor suggestions

Line 39: Therefore, greats challenges – Do you mean greatest? Or great? Please check

It has been modified.

Line 50: with respect.

It has been modified.

Line 66: you are correct. Urea and ammonia. This is not an issue is just a personal question. Did you check creatinine levels? (NOT an issue regarding the manuscript)

It was not measured.

Methods

 The materials and methods section is presented with sufficient detail so that someone can replicate and build on the published results. In addition, the methods were appropriately presented and cited.

Based on your methodology, the players fasted overnight and then had their blood measurements and physical fitness tests one hour before the friendly game. Did they play without eating or drinking anything? I ask this because having a friendly game without consuming anything before the game (after overnight fasting) may have a different effect on metabolic, hormonal and physiological responses instead if they played the game after consuming the recommended carbohydrates etc.

 In line 136 appears the following sentence: “except breakfast after blood samples, which was standardized”.

Discussion

Lines 244-245: Regarding T, previous studies have reported contradictory results at post-match, as T can be decreased [6,7], increased [8] or even unchanged [9]. However, these findings are not relevant according to the aim of this study. So why do you include this if it is irrelevant to your study? I say this because the flow of this paragraph is not very good. A good critical analysis follows. Do not exclude the studies you included that tested T post-match; instead, find a better way to connect it to the subsequent sentences. Change only –however, these findings are not relevant according to the aim of this study- to something more precise.

 It has been modified.

Line 262: may not change respect to pre-match- please check your writing here (with respect instead of respect to pre-match).

It has been modified.

Lines 268-270: “The differences between previous results [9,10] and ours could be explained by two interactive factors: competitive soccer matches increase C levels to a greater magnitude compared to non-competitive fixtures, and male novice players show greater levels of C reactivity compared to high-level soccer players [32]”. You also showed significant increases in C after 72 hours. Which differences are you referring to? Those that found no changes after post-match? (I guess you are not referring to those studies as you did not measure C post-match). Also, are you comparing your results to those that found increases in 24 and 48 h? Still, you did not measure C after 24 and 48 h. I know the study of Ispirlidis et al. (2008) and Silva et al. (2013) that you are citing; therefore, you may want to be more specific here as it is not clear when you say, “The differences between previous results [9,10] and ours could be explained….”

It has been modified.

Line 287: failed to found- check your writing, please.

It has been modified.

Line 294: “enabling precisely information about the metabolically stress experience by soccer players”- check your writing, please.

It has been modified.

Line 311: fatigue may impair jump – change the verb form to impair

It has been modified.

Line 312: results indicate -change the verb form to indicate

It has been modified.

Line 314: which reflects fatigue in a greater extent due inability – please check your writing

It has been modified.

Abstract

  • Lines 16-19: The authors provided the following method in the abstract:

“Before playing a friendly match, blood values of testosterone, cortisol, testosterone to cortisol ratio and urea were collected and squat jump and the Bangsbo Repeated Sprint Ability test were performed. These measurements were considered as baseline (pre-match), and were obtained again after a 72-h recovery period.”

  • This part needs to be rewritten entirely as it is vague and does not provide any information about the participants.

It has been modified.

Introduction

  • Lines 19-20: The exact p-value for the significant changes should be mentioned.

It has been modified.

  • Lines 39-42: Provide 3-5 references for the following sentence:

“Therefore, greats challenges faced by coaches and athletes are to achieve an optimum balance between match-derived biochemical and physiological stress and recovery and to stimulate functional responses which positively affects physical performance.

 References have been added.

  • Line 39: Change the word “greats” to “great”.

It has been modified.

  • Line 41: Change the word “affects” to “affect”.

 It has been modified.

  • The study hypothesis needs to be added at the end of "Introduction".

It has been modified.

Methods

  • Day-to-day test reliability, CV range, and intraclass correlation coefficients for the assessments must be included for ALL the assessments.

 It has been included in the results section (tables and text).

Discussion

  • The "Discussion" section needs to clarify the study's novelty.

It has been added.

  • The conclusions section is too brief.

 It has been expanded.

King Regards

Editor-in-Chief of the Sports:

This letter is in regard to the resubmission (first review) of our manuscript.  As a result of the peer review process in the first round and after considering all of the reviewer comments, we have revised our manuscript according to your suggestions.

We believe that your all comment and our corrections have significantly improved the article, and we hope that it is now suitable for publication in your respected journal.

In Advanced

Best Regards

Julio Calleja González

REVISOR 1

Some minor suggestions

Line 13: Responses are…(minor issue). The singular verb “is” does not appear to agree with the plural subject “responses”.

 It has been modified.

Line 26: You use “functional states” referring to the players' physical performance. The term “functional states” is commonly used in psychology as well as when we refer to brain function. However, it is not used (based on my knowledge) when discussing physical performance. Therefore, I would suggest that you explain it differently.

It has been modified.

Introduction

The introduction highlights the importance of the study. The study's purpose and significance are presented.

Some minor suggestions

Line 39: Therefore, greats challenges – Do you mean greatest? Or great? Please check

It has been modified.

Line 50: with respect.

It has been modified.

Line 66: you are correct. Urea and ammonia. This is not an issue is just a personal question. Did you check creatinine levels? (NOT an issue regarding the manuscript)

It was not measured.

Methods

 The materials and methods section is presented with sufficient detail so that someone can replicate and build on the published results. In addition, the methods were appropriately presented and cited.

Based on your methodology, the players fasted overnight and then had their blood measurements and physical fitness tests one hour before the friendly game. Did they play without eating or drinking anything? I ask this because having a friendly game without consuming anything before the game (after overnight fasting) may have a different effect on metabolic, hormonal and physiological responses instead if they played the game after consuming the recommended carbohydrates etc.

 In line 136 appears the following sentence: “except breakfast after blood samples, which was standardized”.

Discussion

Lines 244-245: Regarding T, previous studies have reported contradictory results at post-match, as T can be decreased [6,7], increased [8] or even unchanged [9]. However, these findings are not relevant according to the aim of this study. So why do you include this if it is irrelevant to your study? I say this because the flow of this paragraph is not very good. A good critical analysis follows. Do not exclude the studies you included that tested T post-match; instead, find a better way to connect it to the subsequent sentences. Change only –however, these findings are not relevant according to the aim of this study- to something more precise.

 It has been modified.

Line 262: may not change respect to pre-match- please check your writing here (with respect instead of respect to pre-match).

It has been modified.

Lines 268-270: “The differences between previous results [9,10] and ours could be explained by two interactive factors: competitive soccer matches increase C levels to a greater magnitude compared to non-competitive fixtures, and male novice players show greater levels of C reactivity compared to high-level soccer players [32]”. You also showed significant increases in C after 72 hours. Which differences are you referring to? Those that found no changes after post-match? (I guess you are not referring to those studies as you did not measure C post-match). Also, are you comparing your results to those that found increases in 24 and 48 h? Still, you did not measure C after 24 and 48 h. I know the study of Ispirlidis et al. (2008) and Silva et al. (2013) that you are citing; therefore, you may want to be more specific here as it is not clear when you say, “The differences between previous results [9,10] and ours could be explained….”

It has been modified.

Line 287: failed to found- check your writing, please.

It has been modified.

Line 294: “enabling precisely information about the metabolically stress experience by soccer players”- check your writing, please.

It has been modified.

Line 311: fatigue may impair jump – change the verb form to impair

It has been modified.

Line 312: results indicate -change the verb form to indicate

It has been modified.

Line 314: which reflects fatigue in a greater extent due inability – please check your writing

It has been modified.

Abstract

  • Lines 16-19: The authors provided the following method in the abstract:

“Before playing a friendly match, blood values of testosterone, cortisol, testosterone to cortisol ratio and urea were collected and squat jump and the Bangsbo Repeated Sprint Ability test were performed. These measurements were considered as baseline (pre-match), and were obtained again after a 72-h recovery period.”

  • This part needs to be rewritten entirely as it is vague and does not provide any information about the participants.

It has been modified.

Introduction

  • Lines 19-20: The exact p-value for the significant changes should be mentioned.

It has been modified.

  • Lines 39-42: Provide 3-5 references for the following sentence:

“Therefore, greats challenges faced by coaches and athletes are to achieve an optimum balance between match-derived biochemical and physiological stress and recovery and to stimulate functional responses which positively affects physical performance.

 References have been added.

  • Line 39: Change the word “greats” to “great”.

It has been modified.

  • Line 41: Change the word “affects” to “affect”.

 It has been modified.

  • The study hypothesis needs to be added at the end of "Introduction".

It has been modified.

Methods

  • Day-to-day test reliability, CV range, and intraclass correlation coefficients for the assessments must be included for ALL the assessments.

 It has been included in the results section (tables and text).

Discussion

  • The "Discussion" section needs to clarify the study's novelty.

It has been added.

  • The conclusions section is too brief.

 It has been expanded.

King Regards

Editor-in-Chief of the Sports:

This letter is in regard to the resubmission (first review) of our manuscript.  As a result of the peer review process in the first round and after considering all of the reviewer comments, we have revised our manuscript according to your suggestions.

We believe that your all comment and our corrections have significantly improved the article, and we hope that it is now suitable for publication in your respected journal.

In Advanced

Best Regards

Julio Calleja González

REVISOR 1

Some minor suggestions

Line 13: Responses are…(minor issue). The singular verb “is” does not appear to agree with the plural subject “responses”.

 It has been modified.

Line 26: You use “functional states” referring to the players' physical performance. The term “functional states” is commonly used in psychology as well as when we refer to brain function. However, it is not used (based on my knowledge) when discussing physical performance. Therefore, I would suggest that you explain it differently.

It has been modified.

Introduction

The introduction highlights the importance of the study. The study's purpose and significance are presented.

Some minor suggestions

Line 39: Therefore, greats challenges – Do you mean greatest? Or great? Please check

It has been modified.

Line 50: with respect.

It has been modified.

Line 66: you are correct. Urea and ammonia. This is not an issue is just a personal question. Did you check creatinine levels? (NOT an issue regarding the manuscript)

It was not measured.

Methods

 The materials and methods section is presented with sufficient detail so that someone can replicate and build on the published results. In addition, the methods were appropriately presented and cited.

Based on your methodology, the players fasted overnight and then had their blood measurements and physical fitness tests one hour before the friendly game. Did they play without eating or drinking anything? I ask this because having a friendly game without consuming anything before the game (after overnight fasting) may have a different effect on metabolic, hormonal and physiological responses instead if they played the game after consuming the recommended carbohydrates etc.

 In line 136 appears the following sentence: “except breakfast after blood samples, which was standardized”.

Discussion

Lines 244-245: Regarding T, previous studies have reported contradictory results at post-match, as T can be decreased [6,7], increased [8] or even unchanged [9]. However, these findings are not relevant according to the aim of this study. So why do you include this if it is irrelevant to your study? I say this because the flow of this paragraph is not very good. A good critical analysis follows. Do not exclude the studies you included that tested T post-match; instead, find a better way to connect it to the subsequent sentences. Change only –however, these findings are not relevant according to the aim of this study- to something more precise.

 It has been modified.

Line 262: may not change respect to pre-match- please check your writing here (with respect instead of respect to pre-match).

It has been modified.

Lines 268-270: “The differences between previous results [9,10] and ours could be explained by two interactive factors: competitive soccer matches increase C levels to a greater magnitude compared to non-competitive fixtures, and male novice players show greater levels of C reactivity compared to high-level soccer players [32]”. You also showed significant increases in C after 72 hours. Which differences are you referring to? Those that found no changes after post-match? (I guess you are not referring to those studies as you did not measure C post-match). Also, are you comparing your results to those that found increases in 24 and 48 h? Still, you did not measure C after 24 and 48 h. I know the study of Ispirlidis et al. (2008) and Silva et al. (2013) that you are citing; therefore, you may want to be more specific here as it is not clear when you say, “The differences between previous results [9,10] and ours could be explained….”

It has been modified.

Line 287: failed to found- check your writing, please.

It has been modified.

Line 294: “enabling precisely information about the metabolically stress experience by soccer players”- check your writing, please.

It has been modified.

Line 311: fatigue may impair jump – change the verb form to impair

It has been modified.

Line 312: results indicate -change the verb form to indicate

It has been modified.

Line 314: which reflects fatigue in a greater extent due inability – please check your writing

It has been modified.

Abstract

  • Lines 16-19: The authors provided the following method in the abstract:

“Before playing a friendly match, blood values of testosterone, cortisol, testosterone to cortisol ratio and urea were collected and squat jump and the Bangsbo Repeated Sprint Ability test were performed. These measurements were considered as baseline (pre-match), and were obtained again after a 72-h recovery period.”

  • This part needs to be rewritten entirely as it is vague and does not provide any information about the participants.

It has been modified.

Introduction

  • Lines 19-20: The exact p-value for the significant changes should be mentioned.

It has been modified.

  • Lines 39-42: Provide 3-5 references for the following sentence:

“Therefore, greats challenges faced by coaches and athletes are to achieve an optimum balance between match-derived biochemical and physiological stress and recovery and to stimulate functional responses which positively affects physical performance.

 References have been added.

  • Line 39: Change the word “greats” to “great”.

It has been modified.

  • Line 41: Change the word “affects” to “affect”.

 It has been modified.

  • The study hypothesis needs to be added at the end of "Introduction".

It has been modified.

Methods

  • Day-to-day test reliability, CV range, and intraclass correlation coefficients for the assessments must be included for ALL the assessments.

 It has been included in the results section (tables and text).

Discussion

  • The "Discussion" section needs to clarify the study's novelty.

It has been added.

  • The conclusions section is too brief.

 It has been expanded.

King Regards

Editor-in-Chief of the Sports:

This letter is in regard to the resubmission (first review) of our manuscript.  As a result of the peer review process in the first round and after considering all of the reviewer comments, we have revised our manuscript according to your suggestions.

We believe that your all comment and our corrections have significantly improved the article, and we hope that it is now suitable for publication in your respected journal.

In Advanced

Best Regards

Julio Calleja González

REVISOR 1

Some minor suggestions

Line 13: Responses are…(minor issue). The singular verb “is” does not appear to agree with the plural subject “responses”.

 It has been modified.

Line 26: You use “functional states” referring to the players' physical performance. The term “functional states” is commonly used in psychology as well as when we refer to brain function. However, it is not used (based on my knowledge) when discussing physical performance. Therefore, I would suggest that you explain it differently.

It has been modified.

Introduction

The introduction highlights the importance of the study. The study's purpose and significance are presented.

Some minor suggestions

Line 39: Therefore, greats challenges – Do you mean greatest? Or great? Please check

It has been modified.

Line 50: with respect.

It has been modified.

Line 66: you are correct. Urea and ammonia. This is not an issue is just a personal question. Did you check creatinine levels? (NOT an issue regarding the manuscript)

It was not measured.

Methods

 The materials and methods section is presented with sufficient detail so that someone can replicate and build on the published results. In addition, the methods were appropriately presented and cited.

Based on your methodology, the players fasted overnight and then had their blood measurements and physical fitness tests one hour before the friendly game. Did they play without eating or drinking anything? I ask this because having a friendly game without consuming anything before the game (after overnight fasting) may have a different effect on metabolic, hormonal and physiological responses instead if they played the game after consuming the recommended carbohydrates etc.

 In line 136 appears the following sentence: “except breakfast after blood samples, which was standardized”.

Discussion

Lines 244-245: Regarding T, previous studies have reported contradictory results at post-match, as T can be decreased [6,7], increased [8] or even unchanged [9]. However, these findings are not relevant according to the aim of this study. So why do you include this if it is irrelevant to your study? I say this because the flow of this paragraph is not very good. A good critical analysis follows. Do not exclude the studies you included that tested T post-match; instead, find a better way to connect it to the subsequent sentences. Change only –however, these findings are not relevant according to the aim of this study- to something more precise.

 It has been modified.

Line 262: may not change respect to pre-match- please check your writing here (with respect instead of respect to pre-match).

It has been modified.

Lines 268-270: “The differences between previous results [9,10] and ours could be explained by two interactive factors: competitive soccer matches increase C levels to a greater magnitude compared to non-competitive fixtures, and male novice players show greater levels of C reactivity compared to high-level soccer players [32]”. You also showed significant increases in C after 72 hours. Which differences are you referring to? Those that found no changes after post-match? (I guess you are not referring to those studies as you did not measure C post-match). Also, are you comparing your results to those that found increases in 24 and 48 h? Still, you did not measure C after 24 and 48 h. I know the study of Ispirlidis et al. (2008) and Silva et al. (2013) that you are citing; therefore, you may want to be more specific here as it is not clear when you say, “The differences between previous results [9,10] and ours could be explained….”

It has been modified.

Line 287: failed to found- check your writing, please.

It has been modified.

Line 294: “enabling precisely information about the metabolically stress experience by soccer players”- check your writing, please.

It has been modified.

Line 311: fatigue may impair jump – change the verb form to impair

It has been modified.

Line 312: results indicate -change the verb form to indicate

It has been modified.

Line 314: which reflects fatigue in a greater extent due inability – please check your writing

It has been modified.

Abstract

  • Lines 16-19: The authors provided the following method in the abstract:

“Before playing a friendly match, blood values of testosterone, cortisol, testosterone to cortisol ratio and urea were collected and squat jump and the Bangsbo Repeated Sprint Ability test were performed. These measurements were considered as baseline (pre-match), and were obtained again after a 72-h recovery period.”

  • This part needs to be rewritten entirely as it is vague and does not provide any information about the participants.

It has been modified.

Introduction

  • Lines 19-20: The exact p-value for the significant changes should be mentioned.

It has been modified.

  • Lines 39-42: Provide 3-5 references for the following sentence:

“Therefore, greats challenges faced by coaches and athletes are to achieve an optimum balance between match-derived biochemical and physiological stress and recovery and to stimulate functional responses which positively affects physical performance.

 References have been added.

  • Line 39: Change the word “greats” to “great”.

It has been modified.

  • Line 41: Change the word “affects” to “affect”.

 It has been modified.

  • The study hypothesis needs to be added at the end of "Introduction".

It has been modified.

Methods

  • Day-to-day test reliability, CV range, and intraclass correlation coefficients for the assessments must be included for ALL the assessments.

 It has been included in the results section (tables and text).

Discussion

  • The "Discussion" section needs to clarify the study's novelty.

It has been added.

  • The conclusions section is too brief.

 It has been expanded.

King Regards

Reviewer 2 Report

Abstract

·         Lines 16-19: The authors provided the following method in the abstract:

“Before playing a friendly match, blood values of testosterone, cortisol, testosterone to cortisol ratio and urea were collected and squat jump and the Bangsbo Repeated Sprint Ability test were performed. These measurements were considered as baseline (pre-match), and were obtained again after a 72-h recovery period.”

·         This part needs to be rewritten entirely as it is vague and does not provide any information about the participants.

Introduction

·         Lines 19-20: The exact p-value for the significant changes should be mentioned.

·         Lines 39-42: Provide 3-5 references for the following sentence:

“Therefore, greats challenges faced by coaches and athletes are to achieve an optimum balance between match-derived biochemical and physiological stress and recovery and to stimulate functional responses which positively affects physical performance.

·         Line 39: Change the word “greats” to “great”.

·         Line 41: Change the word “affects” to “affect”.

·         The study hypothesis needs to be added at the end of "Introduction".

Methods

·         Day-to-day test reliability, CV range, and intraclass correlation coefficients for the assessments must be included for ALL the assessments.

Discussion

·         The "Discussion" section needs to clarify the study's novelty.

·         The conclusions section is too brief.

Author Response

Editor-in-Chief of the Sports:

This letter is in regard to the resubmission (first review) of our manuscript.  As a result of the peer review process in the first round and after considering all of the reviewer comments, we have revised our manuscript according to your suggestions.

We believe that your all comment and our corrections have significantly improved the article, and we hope that it is now suitable for publication in your respected journal.

In Advanced

Best Regards

Julio Calleja González

REVISOR 1

Some minor suggestions

Line 13: Responses are…(minor issue). The singular verb “is” does not appear to agree with the plural subject “responses”.

 It has been modified.

Line 26: You use “functional states” referring to the players' physical performance. The term “functional states” is commonly used in psychology as well as when we refer to brain function. However, it is not used (based on my knowledge) when discussing physical performance. Therefore, I would suggest that you explain it differently.

It has been modified.

Introduction

The introduction highlights the importance of the study. The study's purpose and significance are presented.

Some minor suggestions

Line 39: Therefore, greats challenges – Do you mean greatest? Or great? Please check

It has been modified.

Line 50: with respect.

It has been modified.

Line 66: you are correct. Urea and ammonia. This is not an issue is just a personal question. Did you check creatinine levels? (NOT an issue regarding the manuscript)

It was not measured.

Methods

 The materials and methods section is presented with sufficient detail so that someone can replicate and build on the published results. In addition, the methods were appropriately presented and cited.

Based on your methodology, the players fasted overnight and then had their blood measurements and physical fitness tests one hour before the friendly game. Did they play without eating or drinking anything? I ask this because having a friendly game without consuming anything before the game (after overnight fasting) may have a different effect on metabolic, hormonal and physiological responses instead if they played the game after consuming the recommended carbohydrates etc.

 In line 136 appears the following sentence: “except breakfast after blood samples, which was standardized”.

Discussion

Lines 244-245: Regarding T, previous studies have reported contradictory results at post-match, as T can be decreased [6,7], increased [8] or even unchanged [9]. However, these findings are not relevant according to the aim of this study. So why do you include this if it is irrelevant to your study? I say this because the flow of this paragraph is not very good. A good critical analysis follows. Do not exclude the studies you included that tested T post-match; instead, find a better way to connect it to the subsequent sentences. Change only –however, these findings are not relevant according to the aim of this study- to something more precise.

 It has been modified.

Line 262: may not change respect to pre-match- please check your writing here (with respect instead of respect to pre-match).

It has been modified.

Lines 268-270: “The differences between previous results [9,10] and ours could be explained by two interactive factors: competitive soccer matches increase C levels to a greater magnitude compared to non-competitive fixtures, and male novice players show greater levels of C reactivity compared to high-level soccer players [32]”. You also showed significant increases in C after 72 hours. Which differences are you referring to? Those that found no changes after post-match? (I guess you are not referring to those studies as you did not measure C post-match). Also, are you comparing your results to those that found increases in 24 and 48 h? Still, you did not measure C after 24 and 48 h. I know the study of Ispirlidis et al. (2008) and Silva et al. (2013) that you are citing; therefore, you may want to be more specific here as it is not clear when you say, “The differences between previous results [9,10] and ours could be explained….”

It has been modified.

Line 287: failed to found- check your writing, please.

It has been modified.

Line 294: “enabling precisely information about the metabolically stress experience by soccer players”- check your writing, please.

It has been modified.

Line 311: fatigue may impair jump – change the verb form to impair

It has been modified.

Line 312: results indicate -change the verb form to indicate

It has been modified.

Line 314: which reflects fatigue in a greater extent due inability – please check your writing

It has been modified.

Abstract

  • Lines 16-19: The authors provided the following method in the abstract:

“Before playing a friendly match, blood values of testosterone, cortisol, testosterone to cortisol ratio and urea were collected and squat jump and the Bangsbo Repeated Sprint Ability test were performed. These measurements were considered as baseline (pre-match), and were obtained again after a 72-h recovery period.”

  • This part needs to be rewritten entirely as it is vague and does not provide any information about the participants.

It has been modified.

Introduction

  • Lines 19-20: The exact p-value for the significant changes should be mentioned.

It has been modified.

  • Lines 39-42: Provide 3-5 references for the following sentence:

“Therefore, greats challenges faced by coaches and athletes are to achieve an optimum balance between match-derived biochemical and physiological stress and recovery and to stimulate functional responses which positively affects physical performance.

 References have been added.

  • Line 39: Change the word “greats” to “great”.

It has been modified.

  • Line 41: Change the word “affects” to “affect”.

 It has been modified.

  • The study hypothesis needs to be added at the end of "Introduction".

It has been modified.

Methods

  • Day-to-day test reliability, CV range, and intraclass correlation coefficients for the assessments must be included for ALL the assessments.

 It has been included in the results section (tables and text).

Discussion

  • The "Discussion" section needs to clarify the study's novelty.

It has been added.

  • The conclusions section is too brief.

 It has been expanded.

King Regards